# LoRe: Personalizing LLMs via Low-Rank Reward Modeling

**Avinandan Bose**
University of Washington
FAIR at Meta

**Zhihan Xiong**
University of Washington

**Yuejie Chi**
FAIR at Meta

**Simon Shaolei Du**
University of Washington

**Lin Xiao**
FAIR at Meta

**Maryam Fazel**
University of Washington

Correspondence to: avibose@cs.washington.edu

## Abstract

Personalizing large language models (LLMs) to accommodate diverse user preferences is essential for enhancing alignment and user satisfaction. Traditional reinforcement learning from human feedback (RLHF) approaches often rely on monolithic value representations, limiting their ability to adapt to individual preferences. We introduce a novel framework that leverages *low-rank preference modeling* to efficiently learn and generalize user-specific reward functions. By representing reward functions in a low-dimensional subspace and modeling individual preferences as weighted combinations of shared basis functions, our approach avoids rigid user categorization while enabling scalability and few-shot adaptation. We validate our method on multiple preference datasets, demonstrating superior generalization to unseen users and improved accuracy in preference prediction tasks. The code for our experiments is available at: https://github.com/facebookresearch/LoRe.

## 1 Introduction

Aligning Large Language Models (LLMs) with human values is paramount for enhancing their relatability and effectiveness. Reinforcement Learning from Human Feedback (RLHF) (Christiano et al., 2017) is the standard approach to achieve this alignment. However, conventional approaches often rely on monolithic value representations, which inadequately address the diverse needs of various populations (Bakker et al., 2022; Durmus et al., 2023).

In recent years, there has been a growing advocacy for pluralistic alignment in AI systems. Researchers (Sorensen et al., 2024; Kirk et al., 2024a; Jang et al., 2023) emphasize the importance of designing AI systems that cater to the unique requirements of individuals and groups. This paradigm shift has spurred the development of novel methods, benchmarks, and training datasets. Nevertheless, many existing approaches depend on pre-selected diversity-defining dimensions—such as demographics (Moon et al., 2024; Kwok et al., 2024), personality traits (Castricato et al., 2024; Jiang et al., 2023; Serapio-García et al., 2023; Zhu et al., 2024), and writing styles (Han et al., 2024; Jang et al., 2023; Bai et al., 2022)—which categorize individuals into predefined groups, potentially overlooking intra-group variability. The scarcity of large-scale preference datasets has previously hindered personalized LLM development. However, pioneering efforts by (Kirk et al., 2024b; Zollo et al., 2024) have facilitated the exploration of personalization methods beyond predefined user types.

Early attempts to personalize LLMs involved integrating additional inputs—typically learnable models that generate latent representations of user preferences based on past interactions—into the design of LLMs (Li et al., 2024; Chen et al., 2024b; Woźniak et al., 2024) or reward models (Poddar et al., 2024; Chen et al., 2024a). These strategies, however, often need substantial individual user data or rely on categorizing users based on factors such as

demographics, personalities, etc. To address these limitations, we introduce **LoRe**, a novel *Low-Rank Reward Modeling* framework for few-shot personalization.

LoRe leverages a structured low-rank decomposition of reward functions. This approach allows us to model individual preferences as weighted combinations of the basis reward functions, enabling scalable and statistically efficient adaptation with minimal user-specific data. In contrast to prior approaches, LoRe demonstrates superior generalization capabilities to diverse unseen users while maintaining computational efficiency suitable for real-world deployment. By integrating seamlessly with multi-objective alignment frameworks, LoRe supports personalized response generation without the need for extensive retraining.

## 2 Preliminaries

A crucial step in aligning LLMs through Reinforcement Learning from Human Feedback (RLHF) (Christiano et al., 2017; Ouyang et al., 2022) is learning a reward function that captures human preferences. Unlike traditional supervised fine-tuning, which relies on explicitly labeled data, RLHF enables models to learn from human comparative judgments. This is particularly valuable in settings where direct supervision is impractical, such as optimizing AI systems for subjective qualities like helpfulness or coherence. In practice, human annotators provide feedback by ranking responses to the same prompt, and this data is used to train a reward model that assigns a numerical score to each response.

A common framework for modeling such preferences is the Bradley and Terry (1952) (BT) model. The BT model represents preferences by assigning scalar scores/rewards to items—where an "item" could be any decision, option, or response. Given two items $i$ and $j$ with scores $r_i$ and $r_j$, the probability that item $i$ is preferred over item $j$ follows:

$$\mathbb{P}(i \succ j) = \frac{1}{1 + \exp(-(r_i - r_j))}. \tag{1}$$

In the context of LLMs, the reward function maps prompt-response pairs to a scalar score, indicating response quality. This function, typically represented as $r_\phi : \mathcal{X} \times \mathcal{Y} \to \mathbb{R}$, is trained using human-labeled preference data. Here, $\phi$ is the reward parameterization in the function class $\Phi$. Specifically, for a given prompt $x \in \mathcal{X}$, if human annotators prefer response $y_c \in \mathcal{Y}$ over $y_r \in \mathcal{Y}$, the BT model expresses the probability of this preference as:

$$\mathbb{P}(y_c \succ y_r | x) = \frac{1}{1 + \exp\left(-(r_\phi(x, y_c) - r_\phi(x, y_r))\right)}. \tag{2}$$

Given a dataset $\mathcal{D}$ of pairwise preference feedback consisting of independent samples, where each sample is a triplet $(x, y_c, y_r)$, and $y_c \in \mathcal{Y}$ is the response preferred over $y_r \in \mathcal{Y}$ for the prompt $x \in \mathcal{X}$ drawn uniformly at random. The joint likelihood of the dataset is:

$$\prod_{(x, y_c, y_r) \in \mathcal{D}} \mathbb{P}(y_c \succ y_r | x). \tag{3}$$

Assuming preferences follow the Bradley Terry Model Eq. (1), the parameters of the reward model can be learned by minimizing the negative log-likelihood defined as:

$$\min_{\phi \in \Phi} \sum_{(x, y_c, y_r) \in \mathcal{D}} \log\left(1 + \exp\left(r_\phi(x, y_r) - r_\phi(x, y_c)\right)\right) = \min_{\phi \in \Phi} \sum_{(x, y_c, y_r) \in \mathcal{D}} \ell(r_\phi(x, y_c) - r_\phi(x, y_r)), \tag{4}$$

where $\ell(z) = \log(1 + \exp(-z))$ is the logistic loss function.

## 3 Preference Personalization using LoRe

While the BT model assumes a single underlying reward function shared across users, real-world preferences often exhibit significant variation due to individual experiences, biases, and cultural contexts. Next we present an overview of classical work on collaborative ranking, which provides a way to model diverse user preferences.

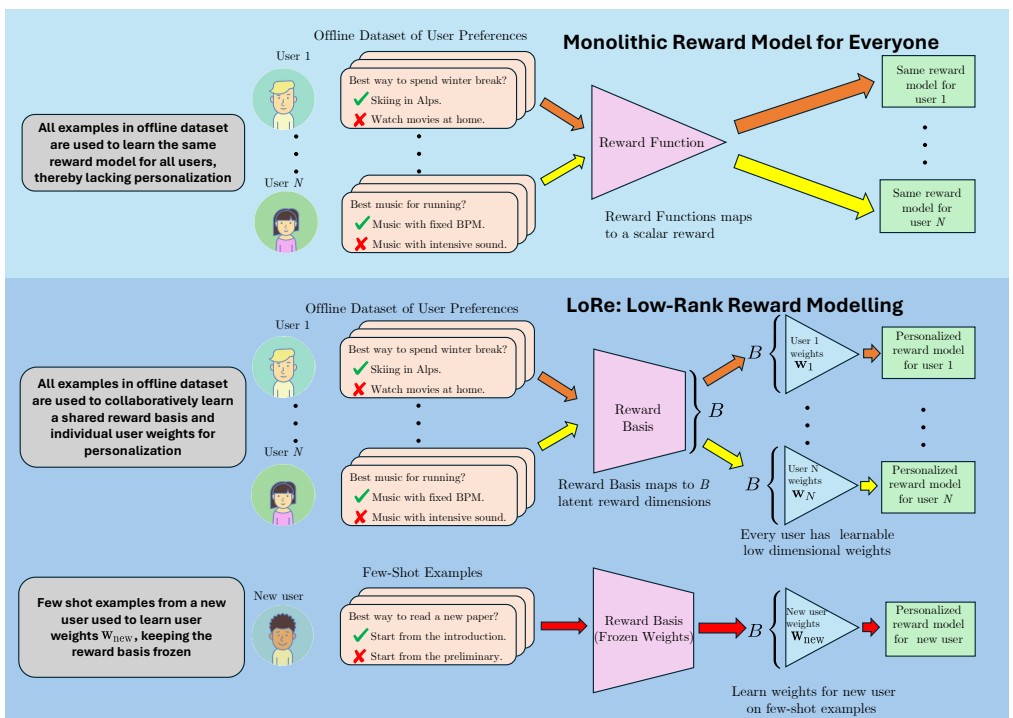

Figure 1: Typically preference data from diverse users is pooled together to train a single reward model for everyone. **LoRe** introduces a more flexible approach by collaboratively learning a shared reward basis from user data. Instead of producing a single reward, this basis generates $B$ latent rewards, which can be combined using a $B$-dimensional weight vector unique to each user to produce personalized rewards. This allows for seamless personalization with minimal effort. For new users, only the user weights need to be learned from few-shot examples, while keeping the reward basis fixed, enabling an efficient and lightweight personalized reward model.

### 3.1 Collaborative Ranking from Pairwise Comparisons

Collaborative ranking (Koren et al., 2009) leverages preference data from multiple users to infer individual preferences across a large item set. Each user provides feedback on only a few item pairs, and the goal is to reconstruct their full preference profile by utilizing shared information across users. This approach accounts for diverse preferences, by avoiding the need to aggregate conflicting opinions into a single reward function. Instead, it models personalized rewards by learning structured representations of user preferences.

Consider $N$ users and $M$ items, where preferences are captured in a matrix $\mathbf{P} \in \mathbb{R}^{N \times M}$. The $i^{\text{th}}$ row of $\mathbf{P}$, denoted $\mathbf{p}_i^\top$, represents user $i$'s rewards across all items. The probability of user $i$ preferring item $c$ over item $r$ is given by the BT model Eq. (1) based on the reward difference $(\mathbf{p}_{i,c} - \mathbf{p}_{i,r})$. Since users provide comparisons for only a small subset of items, recovering the full matrix $\mathbf{P}$ is challenging. A common solution (Lu and Negahban, 2015; Park et al., 2015)—taking into account the similarity among users and items—assumes $\mathbf{P}$ is **low-rank** (has rank $B \ll \min\{M, N\}$) and can be factorized as:

$$\mathbf{P} = \mathbf{WR}, \tag{5}$$

where the rows of $\mathbf{R} \in \mathbb{R}^{B \times M}$ represent a *reward basis*. and $\mathbf{W} \in \mathbb{R}^{N \times B}$ contains user-specific weights. Each user's preference vector is then given by:

$$\mathbf{p}_i = \mathbf{w}_i^\top \mathbf{R}. \tag{6}$$

Here, $\mathbf{w}_i^\top$ determines how user $i$'s preferences combine the basis vectors in $\mathbf{R}$. The objective is to learn this low-rank matrix from a small fraction of observed pairwise comparisons, enabling personalized and scalable reward learning.

## 3.2 Low-Rank Reward Modeling for LLM Alignment

Collaborative ranking, by exploiting the low-rank structure in user preferences, enables few-shot learning on a new user even when only a handful of comparisons are available. The idea is to fix the reward basis $\mathbf{R}$ and learn only a low dimensional weight vector for this user. The main challenge in adapting collaborative ranking to RLHF is the high dimensionality of the item space, which consists of user prompts $x \in \mathcal{X}$ and LLM-generated responses $y \in \mathcal{Y}$, forming items $(x, y) \in \mathcal{X} \times \mathcal{Y}$. User preferences are captured through a limited number of pairwise comparisons $(x, y_c) \succ (x, y_r)$, indicating a preference for $y_c$ over $y_r$ given $x$. However, modern LLMs typically use vocabulary sizes between 32000-128000 ($V$) tokens and support context windows ranging from 2048 to 2 million ($K$) tokens (Touvron et al., 2023; Achiam et al., 2023), the item space scales as $M = V^K$—rendering direct reward basis learning $\mathbf{R} \in \mathbb{R}^{B \times M}$ infeasible.

To address this challenge, we propose a framework that models diverse user preferences through a set of $B$ basis reward functions represented by the Reward Basis $\mathbf{R}_\phi : \mathcal{X} \times \mathcal{Y} \mapsto \mathbb{R}^B$. The individual preference function for user $i$, $\mathbf{p}_i : \mathcal{X} \times \mathcal{Y} \to \mathbb{R}$, is defined as:

$$\mathbf{p}_i := \mathbf{w}_i^\top \mathbf{R}_\phi, \tag{7}$$

where $\mathbf{w}_i^\top \in \Delta^{B-1}$ is a normalized weight vector inherent to the user. By leveraging a straightforward low-rank matrix factorization, we efficiently capture diverse user preferences. This approach, overlooked in favor of more complex methods in prior work, highlights the strength of collaborative ranking across diverse applications. The simplicity of our method is a major advantage, enabling significant performance gains (see Sec. 5) while being easy to integrate with various downstream tasks (see Sec. 3.4). The space $\Phi$ of the learnable parameters depends on the use case and the sample size available for training. We provide a few examples below.

- **Example 1.** Fine-tuning only the final layer of a pre-trained reward model (Ziegler et al., 2019): Standard transformer-based models output a single scalar reward. To modify the reward model to output a $B$-dimensional representation, we learn a simple linear transformation on top of the embeddings generated by the pre-final layer, denoted as $\mathbf{e} : \mathcal{X} \times \mathcal{Y} \mapsto \mathbb{R}^D$, while keeping other layers frozen. The reward basis is then defined as:

$$\mathbf{R}_\phi(x, y) = \mathbf{A}\mathbf{e}(x, y), \tag{8}$$

  where $\mathbf{A} \in \mathbb{R}^{B \times D}$ is a learnable matrix that projects $\mathbf{e}(x, y)$ into a $B$-dimensional space.

- **Example 2.** Similar to Example 1, we modify the pre-trained reward model to output a $B$-dimensional representation by applying a learnable transformation to the embeddings from the pre-final layer. Instead of using a linear transformation, we train a shallow multi-layer perceptron (MLP) $f_\phi$ on top of these frozen embeddings (Wang et al., 2024):

$$\mathbf{R}_\phi(x, y) = f_\phi(\mathbf{e}(x, y)), \tag{9}$$

  where $\phi$ are the parameters of the MLP. This allows for more expressive transformations while keeping the earlier layers of the reward model frozen.

- **Example 3.** Fine-tuning earlier layers with LoRA (Low-Rank Adaptation) (Hu et al., 2021): Instead of keeping the earlier layers frozen, we can allow them to be fine-tuned in a parameter-efficient way using LoRA. As in the previous examples, we first modify the pre-trained reward model to output a $B$-dimensional representation by applying a learnable transformation to the embeddings from the pre-final layer. However, in addition to learning a transformation on top of these embeddings, we also fine-tune the transformer's earlier layers by introducing low-rank adaptation matrices.

### 3.3 The LoRe Workflow

In this section, we outline the LoRe workflow in full generality. The specific choices made for our experiments are discussed in Section 5.

**Collecting User Preference Data:** We assume access to preference feedback data from **seen users**, denoted by the set $\mathcal{U}_{\text{seen}}$. Each user provides a set of labeled pairs of responses, denoted as $\mathcal{D}_i = \{(x, y_c, y_r)\}$, where $x$ is the input, and $y_c$ and $y_r$ are the chosen and rejected options, respectively. The training dataset $\mathcal{D}_{\text{train}}$ is the collection of $\mathcal{D}_i$ of all users in $\mathcal{U}_{\text{seen}}$.

**Jointly Learning Basis Reward and User Preference Weights:** Assuming that the preference for each user follows the BT Model with their personalized reward function $\mathbf{p}_i$, the learning problem boils down to the following maximum likelihood estimation problem:

$$\min_{\phi:\Phi,\{\mathbf{w}_i \in \Delta^{B-1}\}_{i \in \mathcal{U}_{\text{seen}}}} \sum_{i \in \mathcal{U}_{\text{seen}}} \frac{1}{|\mathcal{D}_i|} \sum_{(x,y_c,y_r) \in \mathcal{D}_i} \ell(\mathbf{w}_i^\top (\mathbf{R}_\phi(x, y_c) - \mathbf{R}_\phi(x, y_r))), \qquad (10)$$

where $\ell(\cdot)$ is the logistic loss function as described in Eq. (4).

Once the learner has learned these parameters, they are interested in generalizing to new prompt queries by the users in $\mathcal{U}_{\text{seen}}$, as well as being able to adapt to preferences of new users denoted by the set $\mathcal{U}_{\text{unseen}}$.

**Few Shot Learning for New Users (Unseen User Generalization):** This type of generalization involves predicting well for users whose preference data was not part of the training data at all, i.e., completely new users with unseen preferences. These users are termed as **unseen users** denoted by the set $\mathcal{U}_{\text{unseen}}$. These users have few interaction data points, that is termed as $\mathcal{D}_{\text{fewshot}}$, and this is used to few-shot learn these users' preferences $\{\mathbf{w}_i\}_{i \in \mathcal{U}_{\text{unseen}}}$ by optimizing Eq. (11).

For any user $i \in \mathcal{U}_{\text{unseen}}$ with few feedback samples denoted by $\mathcal{D}_i \in \mathcal{D}_{\text{fewshot}}$, we estimate their preference $\mathbf{w}_{\text{new}} \in \Delta^{B-1}$, keeping the reward basis $\mathbf{R}_\phi$ fixed from Eq. (10) as follows:

$$\mathbf{w}_{\text{new}} = \operatorname*{argmin}_{\mathbf{w} \in \Delta^{B-1}} \sum_{(x,y_c,y_r) \in \mathcal{D}_i} \ell(\mathbf{w}^\top (\mathbf{R}_\phi(x, y_c) - \mathbf{R}_\phi(x, y_r))). \qquad (11)$$

### 3.4 Personalized Response Generation via Steerable Multi-Objective Alignment

Recently, there has been a growing interest in Multi-Objective Alignment (MOA) in LLMs. Formally, let $\mathbf{R}_\phi : \mathcal{X} \times \mathcal{Y} \to \mathbb{R}^B$ represent $B$ reward functions for different (often conflicting) objectives. For instance, one objective may favor detailed explanations, while another prioritizes conciseness. The goal is to generate responses with varying emphasis, dictated by a weight vector $\mathbf{w} \in \mathbb{R}^B$, yielding the reward function $\mathbf{w}^\top \mathbf{R}_\phi$. A naive approach would be to train a language model for every possible $\mathbf{w}$, but this becomes infeasible as the space of $\mathbf{w}$ is infinite.

Prior work (Wu et al., 2023; Zhou et al., 2023; Rame et al., 2024; Jang et al., 2023; Shi et al., 2024) demonstrates that it is possible to learn only $B$ language models corresponding to these $B$ reward functions and still generate responses for any arbitrary weight $\mathbf{w}$ at inference time. However, *these works assume known reward models* (using off-the-shelf reward models for observable objectives like harmlessness, conciseness, etc.) and *provided preference weights*.

In personalization, objectives are often latent and subjective, requiring learning rather than explicit specification. Furthermore, preferences are inherently subjective and can be difficult to articulate, making it challenging for users to specify precisely what weights they want. Modeling personalized rewards as linear combinations of basis functions enables adaptation to users' implicit preferences. Our research is *orthogonal yet complementary* to this line of work, enabling seamless integration into advances in multi-objective alignment.

# 4 Related Work and Contributions

**Personalized Reward Learning from Human Feedback:** This line of research typically captures diversity in user preferences in one of the following ways: 1.**Explicitly categorize users** based on observable traits (e.g., demographics, personality traits) and train separate reward models for each category (Jiang et al., 2023; Zhu et al., 2024; Bose et al., 2024). This approach is inherently limited in granularity and struggles with cases where user preferences do not align neatly with predefined categories. 2. **Learn per-user reward models** by conditioning on latent representations of user preferences (Poddar et al., 2024; Chen et al., 2024a; Lee et al., 2024). These methods require significant data for each user and do not generalize well to new users with limited feedback. We elaborate on two of these methods next.

**Personalized reward modeling for pluralistic alignment (PAL) :** (Chen et al., 2024a) assume that the latent preference of each user is modeled by an unknown ideal point $\mathbf{z}_i \in \mathbb{R}^D$. They propose two methods to represent these ideal points:

**PAL-A:** In this approach, the ideal points factorize as $\mathbf{z}_i = \mathbf{Q}\mathbf{w}_i$, where $\mathbf{Q} \in \mathbb{R}^{D \times B}$ represents $B$ prototype ideal points and $\mathbf{w}_i \in \Delta^{B-1}$ represents weights over these prototypes for user $i$. Given a pre-trained embedding function $\mathbf{e} : \mathcal{X} \times \mathcal{Y} \mapsto \mathbb{R}^D$, the reward for user $i$ for a prompt response pair $(x, y) \in \mathcal{X} \times \mathcal{Y}$ is given by the Euclidean distance square between $\mathbf{e}(x, y)$ and $\mathbf{Q}\mathbf{w}_i$ in a learnable representation space defined by function $f_\theta : \mathbb{R}^D \mapsto \mathbb{R}^d$ (a shallow MLP with parameters $\theta$):

$$\mathbf{p}_i(x, y) := \|f_\theta(\mathbf{e}(x, y)) - f_\theta(\mathbf{Q}\mathbf{w}_i)\|_2^2. \tag{12}$$

The learnable parameters are $\theta, \mathbf{Q}, \{\mathbf{w}_i\}_{i \in \mathcal{U}_{\text{seen}}}$. This method aims to generalize to unseen users by covering the user space with the prototype matrix.

**PAL-B:** Here, the user ideal point is a function of the prompt, expressed as $\mathbf{z}_i(x) = \mathbf{G}_\phi(\mathbf{e}(x))\mathbf{w}_i$. The personalized reward function is defined as:

$$\mathbf{p}_i(x, y) := \mathbf{z}_i(x)^\top f_\theta(\mathbf{e}(y)) = \mathbf{w}_i^\top \mathbf{G}_\phi^\top(\mathbf{e}(x)) f_\theta(\mathbf{e}(y)), \tag{13}$$

where $\mathbf{G}_\phi : \mathbb{R}^D \mapsto \mathbb{R}^d \times \mathbb{R}^K$ and $f_\theta : \mathbb{R}^D \mapsto \mathbb{R}^d$. Mathematically, PAL-B is a special case of Eq. (7) where $\mathbf{R}_\phi(x, y) := \mathbf{G}_\psi^\top(\mathbf{e}(x)) f_\theta(\mathbf{e}(y))$. This decomposition of $\mathbf{R}_\phi$ is not novel to PAL-B and has already appeared in a prior work (Wang et al., 2024).

**Variational Preference Learning (VPL) (Poddar et al., 2024):** The latent preference of each user is denoted by $\mathbf{z}_i \in \mathbb{R}^D$, which is the output of an encoder function $\mathbf{Q}_\theta : \mathcal{D}_i \mapsto \mathbb{R}^D$, that maps the user preference data $\mathcal{D}_i$ to a latent code. Given a pre-trained embedding function $\mathbf{e} : \mathcal{X} \times \mathcal{Y} \mapsto \mathbb{R}^D$, the reward for user $i$ for a prompt response pair $(x, y) \in \mathcal{X} \times \mathcal{Y}$ is given by a learnable function $R_\phi : \mathbb{R}^{2D} \mapsto \mathbb{R}$, as:

$$\mathbf{p}_i(x, y) := \mathbf{R}_\phi(\mathbf{e}(x, y); \mathbf{Q}_\theta(\mathcal{D}_i)). \tag{14}$$

The noise in the latent space encourages the reward model to learn over the entire latent space, which encourages the production of meaningful rewards for unseen users, with the learnable parameters being those of $\mathbf{R}_\phi$ and $\mathbf{Q}_\theta$.

**Personalized Response Generation:** Following the idea of Direct Preference Optimization (DPO) (Rafailov et al., 2024), which learns a response generation policy (a language model) without explicitly learning a reward function, recent approaches directly model personalized responses. Li et al. (2024) use an encoder to generate latent user embeddings, conditioning a language model on them via DPO. Chen et al. (2024b) personalizes responses through user-specified prompts, while Woźniak et al. (2024) incorporates user information as input features. Other works explore limited personalization settings, such as multiple-choice questions (Zhao et al., 2023), explicit human corrections (Shaikh et al., 2024), and few-shot adaptation with synthetic users (Singh et al., 2025). In Appendix A, we show that the core idea of LoRe naturally extends to response generation without explicit reward learning too.

Evaluating personalized response generation is particularly challenging. Unlike reward learning, where user data can be held out for validation, there is no access to users in offline datasets to label their preferences on newly-generated responses. Instead, evaluations rely on heuristic LLM-based evaluators, which are typically trained using monolithic Bradley-Terry reward models assuming a single global ranking of responses, failing to capture diverse user preferences. Hence, as discussed in Panickssery et al. (2024); Dong et al. (2024), these models reinforce bias; favoring dominant preference patterns while undervaluing minority preferences and struggle with ambiguous or tied preferences, leading to systematic misalignment with real user satisfaction.

**Contributions** We state our contributions and how we overcome limitations in prior work:

1. **Latent Basis Reward Functions for Personalized Alignment** LoRe introduces a structured approach to personalization by learning a set of primitive reward functions (basis functions) that span the space of individual reward models. Each user's preference is represented as a weighted combination of these basis functions, enabling smooth adaptation without requiring predefined user categories or extensive per-user data.
2. **Decoupled Learning for Efficient and Generalizable Adaptation** LoRe separates the learning of basis reward functions from user-specific weights, enabling rapid few-shot personalization. Once the basis functions are learned, a new user's preferences can be captured with only a small number of interactions, making it practical for real-world deployment. By capturing the underlying structure of user preferences, LoRe generalizes effectively to unseen users with minimal data. Unlike latent-code-based methods like VPL or PAL that require separate modules for inferring user representations, LoRe directly learns a compact basis, reducing the number of learnable parameters and improving both efficiency and generalization.
3. **Scalability to Large and Diverse User Populations** Unlike approaches that either assume homogeneous reward models (BT) or struggle with scalability (PAL, VPL), LoRe maintains strong performance on large-scale personalization tasks. The low-rank decomposition reduces computational overhead while preserving expressiveness (cf. Sec. 5).
4. **Integration with Multi-Objective Alignment for Response Generation** LoRe naturally extends to personalized response generation by leveraging its basis reward functions. Unlike PAL and VPL, which require additional policy networks to generate responses, our method integrates seamlessly with steerable multi-objective alignment frameworks.
5. **Bridging the Gap Between Explicit Categorization and Per-User Models** Many personalization methods either cluster users into predefined categories (demographics, personality types) or train separate models for each user. LoRe avoids these extremes by learning a flexible, data-efficient representation of user preferences that adapts without requiring extensive individual data.

By combining structured reward decomposition, scalable adaptation, and efficient integration with response generation, LoRe offers a principled and practical approach to personalized RLHF, addressing key limitations of prior methods while enabling new capabilities.

## 5 Experiments

**Evaluation Metrics:** We evaluate the reward model's accuracy on unseen response pairs for both seen and unseen users:

$$\frac{1}{|\tilde{\mathcal{D}}_i|} \sum_{(x,y_c,y_r) \in \tilde{\mathcal{D}}_i} \mathbb{I}[\mathbf{w}_i^\top (\mathbf{R}_\phi(x, y_c) - \mathbf{R}_\phi(x, y_r)) > 0], \tag{15}$$

where $\mathbb{I}[\cdot]$ is the indicator function, equal to 1 if the condition holds and 0 otherwise.

We define four dataset splits: $\mathcal{D}_{\text{train}}$ contains labeled preference data from seen users $\mathcal{U}_{\text{seen}}$ used to train the reward basis, $\mathcal{D}_{\text{test}}^{\text{seen}}$ evaluates generalization to new response pairs for seen users, $\mathcal{D}_{\text{fewshot}}$ provides a small set of labeled data for unseen users $\mathcal{U}_{\text{unseen}}$, and $\mathcal{D}_{\text{test}}^{\text{unseen}}$ assesses generalization to new users. We evaluate:

1. **Seen Accuracy:** Generalization to new response pairs for seen users using $\mathcal{D}_{\text{test}}^{\text{seen}}$.

| Setting | PersonalLLM | | | | | | | | |
|---|---|---|---|---|---|---|---|---|---|
| | Very Diverse ($\alpha = 0.001$) | | | Moderately Diverse ($\alpha = 0.01$) | | | Near Uniform ($\alpha = 0.1$) | | |
| Method | Seen | Unseen | Overall | Seen | Unseen | Overall | Seen | Unseen | Overall |
| ref | 78.4 ± 1.2 | 76.1 ± 1.5 | 77.3 ± 1.3 | 78.1 ± 1.4 | 77.8 ± 1.6 | 77.9 ± 1.5 | 82.7 ± 1.0 | 83.7 ± 1.1 | 83.2 ± 1.0 |
| BT | 86.3 ± 1.0 | 86.4 ± 1.3 | 86.4 ± 1.1 | 87.8 ± 1.2 | 87.2 ± 1.4 | 87.5 ± 1.3 | 93.2 ± 0.9 | 93.1 ± 1.0 | 93.2 ± 0.9 |
| VPL | 86.4 ± 1.3 | 86.5 ± 1.2 | 86.5 ± 1.3 | 93.9 ± 1.1 | 84.1 ± 1.5 | 89.0 ± 1.3 | 92.0 ± 1.0 | 92.8 ± 0.8 | 92.4 ± 0.9 |
| PAL | 85.0 ± 1.3 | 86.5 ± 1.2 | 85.7 ± 1.3 | 86.1 ± 1.1 | 87.1 ± 1.5 | 86.6 ± 1.3 | 91.7 ± 1.0 | 91.8 ± 0.8 | 91.7 ± 0.9 |
| LoRe | **94.3** ± 0.9 | **93.3** ± 1.0 | **93.8** ± 0.9 | **94.6** ± 0.8 | **93.6** ± 1.1 | **94.1** ± 0.9 | **96.0** ± 0.7 | **96.1** ± 0.8 | **96.0** ± 0.7 |

Table 1: Using **PersonalLLM** we generate 1000 seen users and 1000 unseen users. In particular, we use 45 examples per seen user and 9 few-shot examples per unseen user.

2. **Unseen Accuracy:** Generalization to new response pairs for unseen users on $\mathcal{D}_{\text{test}}^{\text{unseen}}$, where user preferences are learned from few-shot examples in $\mathcal{D}_{\text{fewshot}}$.

3. **Few-shot Ability:** Accuracy on $\mathcal{D}_{\text{test}}^{\text{unseen}}$ upon varying the number of examples in $\mathcal{D}_{\text{fewshot}}$.

**Baselines:** We use a pre-trained reward model (Liu et al., 2024) to generate fixed embeddings $\mathbf{e}(x, y) \in \mathbb{R}^D$, where $D = 4096$. We compare against 1) **Reference Model** (Liu et al., 2024), 2) **BT** (monolithic reward model, applying a learnable linear mapping from fixed embeddings $\mathbf{e}(x, y)$ to a scalar reward), 3) **VPL** (Poddar et al., 2024), 4) **PAL** (Chen et al., 2024a) (both using their respective architectures over the same fixed embeddings $\mathbf{e}(x, y)$), and 5) **LoRe** (applying a learnable linear transformation on $\mathbf{e}(x, y)$ to map embeddings to $\mathbb{R}^B$ (corresponding to Example 1 in Sec. 3.2). A detailed description of all baselines is presented in Appendix B.1.

**Semi-synthetic Preference Dataset:** The PersonalLLM dataset (Zollo et al., 2024) contains 10,402 prompts, each with responses from eight top LLMs (e.g., GPT-4o, Claude 3 Opus, Mixtral8x22B). Each response is scored by 10 reward models from Reward Bench, built on popular base models such as Llama3, Mistral, and Gemma, with diverse preferences. Given a prompt $x$ and response $y$, the reward vector is $\mathbf{R}(x, y) \in \mathbb{R}^{10}$. The dataset is split into 9,402 training and 1,000 test prompts.

Synthetic users are generated by sampling a preference vector $\mathbf{w} \sim \text{Dirichlet}(\alpha)$ and computing response scores as $\mathbf{w}^\top \mathbf{R}(x, y)$. We vary $\alpha$ in the range $\{0.1, 0.01, 0.001\}$, where a larger $\alpha$ results in more uniform preferences and a smaller $\alpha$ leads to more discrete user types. We then categorize users as follows: 1. **Very Diverse** ($\alpha = 0.001$): Aligning closely with one of the 10 reward models. 2. **Moderately diverse** ($\alpha = 0.01$): A balance between specific and broad preferences. 3. **Near Uniform** ($\alpha = 0.1$): The most uniform preferences.

For each user, we store the highest/lowest scored responses and simulate 1000 seen and 1000 unseen users. Each seen user gets 45 prompts from the training set $\mathcal{D}_{\text{train}}$, while each unseen user gets 9 prompts to form $\mathcal{D}_{\text{fewshot}}$. All 1000 test prompts form $\mathcal{D}_{\text{test}}^{\text{seen}}, \mathcal{D}_{\text{test}}^{\text{unseen}}$ to test the performance of the learnt models.

**Summarization Task on Real Users:** We use the TLDR dataset, where each comparison consists of a Reddit post, two summaries, and the worker ID who annotated it (Stiennon et al., 2020). After filtering out workers with fewer than 50 annotations, we retain 40 workers. They are split into two equal groups of 20, corresponding to $\mathcal{U}_{\text{seen}}$ and $\mathcal{U}_{\text{unseen}}$.

| Setting | Reddit TLDR | | | | | |
|---|---|---|---|---|---|---|
| | 100 examples per seen user | | | 150 examples per seen user | | |
| Method | Seen | Unseen | Overall | Seen | Unseen | Overall |
| ref | 56.3 ± 1.3 | 57.3 ± 1.4 | 56.8 ± 1.2 | 56.3 ± 1.2 | 57.3 ± 1.5 | 56.8 ± 1.3 |
| BT | 60.0 ± 1.4 | 60.0 ± 1.2 | 60.0 ± 1.3 | 63.2 ± 1.1 | 64.3 ± 1.3 | 63.7 ± 1.2 |
| VPL | 63.6 ± 1.3 | 62.1 ± 1.4 | 62.9 ± 1.2 | 63.4 ± 1.3 | 62.7 ± 1.2 | 63.1 ± 1.3 |
| PAL | 64.1 ± 1.1 | 64.9 ± 1.5 | 64.5 ± 1.2 | 64.4 ± 1.3 | 63.8 ± 1.2 | 64.1 ± 1.3 |
| LoRe | **65.0** ± 1.1 | **66.2** ± 1.2 | **65.6** ± 1.1 | **66.2** ± 1.0 | **66.7** ± 1.1 | **66.5** ± 1.0 |

Table 2: We split **Reddit TLDR** into 20 seen and 20 unseen users, with 50 few-shot examples per unseen user. We vary the number of examples per seen user to learn the reward basis.

Each worker has an average of 4,000 labeled pairs, which are evenly divided into a training set and a test set. To construct $\mathcal{D}_{\text{train}}$, we randomly sample $\{100, 150\}$ pairs from each seen user's training data. For unseen users, we randomly select 50 examples from their training data to form $\mathcal{D}_{\text{fewshot}}$. Evaluation is conducted on the full set of labeled test samples, corresponding to $\mathcal{D}_{\text{test}}^{\text{seen}}$ and $\mathcal{D}_{\text{test}}^{\text{unseen}}$.

**Real-World Preference Dataset with a Large and Diverse User Base** The PRISM dataset (Kirk et al., 2024b) is a comprehensive resource for LLM feedback analysis, featuring 1,500 participants from 75 countries. It provides fine-grained feedback on both contextual and stated preferences, collected from 8,011 live conversations across 21 different LLMs.

After filtering out users with fewer than six dialogues, we randomly split the remaining participants into two equal groups, resulting in $|\mathcal{U}_{\text{seen}}| = |\mathcal{U}_{\text{unseen}}| = 643$ users, with each user averaging seven dialogues. For each user, half of their interactions are used for training ($\mathcal{D}_{\text{train}}, \mathcal{D}_{\text{fewshot}}$), which informs reward model learning and the preferences of unseen users. The remaining interactions are reserved for evaluation ($\mathcal{D}_{\text{test}}^{\text{seen}}, \mathcal{D}_{\text{test}}^{\text{unseen}}$).

We repeat each experiment 20 times with different random splits, reporting the mean and standard deviation in Tables 1, 2, and 3. To further analyze the scalability of the few-shot adaptation phase, we vary the number of few-shot samples and plot the average performance across 20 runs in Figure 2.

| Method | PRISM | | |
|---|---|---|---|
| | Seen | Unseen | Overall |
| ref | $58.8 \pm 1.1$ | $57.3 \pm 1.2$ | $58.0 \pm 1.0$ |
| BT | $64.0 \pm 1.0$ | $61.0 \pm 1.2$ | $62.5 \pm 1.1$ |
| VPL | $64.6 \pm 1.0$ | $58.2 \pm 1.2$ | $61.4 \pm 1.1$ |
| PAL | $70.8 \pm 1.0$ | $59.0 \pm 1.2$ | $64.92 \pm 1.1$ |
| LoRe | $\mathbf{71.0} \pm 0.9$ | $\mathbf{71.0} \pm 1.0$ | $\mathbf{71.0} \pm 0.8$ |

Table 3: We split **PRISM** into 643 seen and 643 unseen users. On average there are only 3.84 examples per seen and 3.87 examples per unseen user, making generalization to unseen users challenging for baselines.

**Analysis** In the PersonalLLM dataset, as diversity decreases, all methods improve, but LoRe consistently achieves the best performance across all settings. VPL and PAL, which are personalization baselines, struggle to remain competitive when the number of users is large, as seen in both PersonalLLM and PRISM. Their performance is often close to BT, which does not personalize at all, indicating a lack of scalability. However, when the number of users is smaller, as in the Reddit TLDR dataset, both VPL and PAL perform well, further reinforcing their limitations in scaling to larger personalization tasks. Additionally, VPL exhibits signs of overfitting, performing well on seen users but significantly worse on unseen users, highlighting its inability to generalize effectively. In contrast, LoRe consistently outperforms all other methods, demonstrating strong scalability, generalization, and adaptability across varying levels of personalization diversity.

We also analyze the dependence of few-shot examples on unseen accuracy in Figure 2. The BT and ref models are not capable of personalization to new users and hence demonstrate the same performance throughout. We observe that VPL and PAL do not change their performance much either as number of few shot examples increase. We found that the high-dimensional latent codes do not change significantly for both VPL and PAL as the number of few-shot examples increased. For LoRe, the unseen accuracy steadily increases across all datasets, as the number of examples increases.

## 6   Conclusion and Future Work

We introduced LoRe, a novel framework for personalizing LLMs via Low-Rank Reward Modeling. Our approach improves RLHF personalization by leveraging a structured decomposition of reward functions, enabling efficient adaptation to diverse user preferences with minimal data. Extensive evaluations demonstrated LoRe's superior generalization to seen and unseen users while maintaining scalability and efficiency. Compared to baseline methods, LoRe consistently achieved better unseen user adaptation and preference prediction accuracy. It remains effective even as the number of users increases, breaking a key limitation in prior work. Future directions include extending LoRe to online RLHF with explorative data collection.

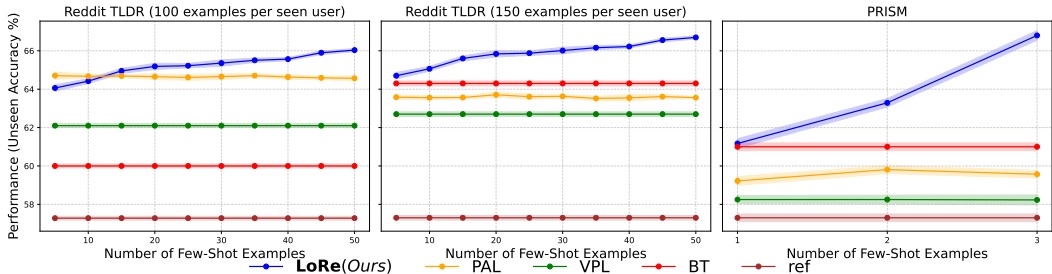

Figure 2: We vary the number of few-shot samples and repeat each experiment 20 times, randomly subsampling different examples in each run. The plot reports the average performance (unseen accuracy) along with standard deviations. Notably, VPL, which infers the latent code from few-shot examples without the ability to relearn it, shows limited improvement as the number of examples increases. While PAL exhibits some gains, our algorithm's performance improves significantly faster in comparison.

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

## A  LoRe directly for language generation

A language model is a policy network that assigns the likelihood of a response $y \in \mathcal{Y}$, given prompt $x \in \mathcal{X}$ denoted by $\pi_\theta : \mathcal{X} \mapsto \Delta^{|\mathcal{Y}|-1}$. Given a reward function $r_\phi : \mathcal{X} \times \mathcal{Y} \mapsto \mathbb{R}$ and a reference policy $\pi_{\text{ref}}$ (typically obtained by supervised finetuning a pre-trained model), the objective is to learn a policy $\pi_\theta$ that maximizes the KL-regularized reward maximization problem:

$$\theta^* = \arg\max_{\theta \in \Theta} \mathbb{E}_{x \sim \mathcal{D}, y \sim \pi_\theta(\cdot|x)}[r_\phi(x, y) - \beta \mathbb{D}_{\text{KL}}(\pi_\theta(y|x)||\pi_{\text{ref}}(y|x)]. \tag{16}$$

Rafailov et al. (2024) shows that the optimal policy for the KL-regularized reward maximization problem satisfies

$$\pi_{\theta^*}(y|x) = \frac{\pi_{\text{ref}}(y|x)}{Z_\phi(x)} \exp\left(\frac{r_\phi(x, y)}{\beta}\right) \implies r_\phi(x, y) = \beta \log \frac{\pi_{\theta^*}(y|x)}{\pi_{\text{ref}}(y|x)} + \beta Z_\phi(x), \tag{17}$$

where $Z_\phi(x)$ is a normalization factor. Since there is a direct correspondence between the reward function $r_\phi$ and the optimal policy that maximizes the KL-regularized reward, Rafailov et al. (2024) proposes Direct Preference Optimization (DPO), which directly learns the policy $\pi_{\theta^*}$ without first learning reward model $r_\phi$ on offline preference data. Then, by using this closed-form expression for $r_\phi(x, y)$ and applying minimizing negative log-likelihood to the Bradley-Terry model, we can obtain

$$\min_{\phi \in \Phi} \sum_{(x, y_c, y_r) \in \mathcal{D}} \ell(r_\phi(x, y_c) - r_\phi(x, y_r))$$

$$= \min_{\theta \in \Theta} \sum_{(x, y_c, y_r) \in \mathcal{D}} \ell\left(\beta \log \frac{\pi_\theta(y_c|x)}{\pi_{\text{ref}}(y_c|x)} - \beta \log \frac{\pi_\theta(y_r|x)}{\pi_{\text{ref}}(y_r|x)}\right) \qquad \text{(by Eq. (17).)}$$

**Learning a Policy Basis:** Instead of first learning a reward basis function $\mathbf{R}_\phi : \mathcal{X} \times \mathcal{Y} \mapsto \mathbb{R}^B$, it is equivalent to learning (fine-tuning) $B$ basis policies $\{\pi_{\theta_i} : \mathcal{X} \mapsto \Delta^{|\mathcal{Y}|-1}\}_{i \in [B]}$, which produces $B$ likelihoods, each corresponding to the optimal policy for each of the latent reward dimensions.

Conditioned on a prompt $x$, and a pair of responses $(y, \tilde{y})$, the probability of a user with preference weights $\mathbf{w} \in \Delta^{B-1}$ of choosing $y$ over $\tilde{y}$ under the Bradley-Terry model can be written as:

$$\mathbb{P}[y \succ \tilde{y}|x] = \frac{1}{1 + \exp\left(-\mathbf{w}^\top \left(\mathbf{R}_\phi(x, y) - \mathbf{R}_\phi(x, \tilde{y})\right)\right)}$$

$$= \frac{1}{1 + \exp\left(-\beta \sum_{j \in [B]} w^{(j)} \left(\log \frac{\pi_{\theta_j}(y|x)}{\pi_{\text{ref}}(y|x)} - \log \frac{\pi_{\theta_j}(\tilde{y}|x)}{\pi_{\text{ref}}(\tilde{y}|x)}\right)\right)}, \qquad \text{(by Eq. (17).)}$$

where $w^{(j)}$ is the $j^{\text{th}}$ entry of the vector $\mathbf{w}$. Analogous to Eq. (10), given a dataset of offline preferences, the parameters $\theta$ and $\{\mathbf{w}_i\}_{i \in \mathcal{U}_{\text{seen}}}$ can be learnt by plugging in the negative log-likelihood loss as

$$\min_{\{\theta_j : \Theta\}_{j \in [B]}, \{\mathbf{w}_i \in \Delta^{B-1}\}_{i \in \mathcal{U}_{\text{seen}}}} \sum_{i \in \mathcal{U}_{\text{seen}}} \frac{1}{|\mathcal{D}_i|} \sum_{(x, y_c, y_r) \in \mathcal{D}_i} \ell\left(\sum_{j \in [B]} w_i^{(j)} \left(\beta \log \frac{\pi_{\theta_j}(y_c|x)}{\pi_{\text{ref}}(y_c|x)} - \beta \log \frac{\pi_{\theta_j}(y_r|x)}{\pi_{\text{ref}}(y_r|x)}\right)\right). \tag{18}$$

For a new user, $\theta$ is frozen, and only the user weight $\mathbf{w}_{\text{new}}$ is estimated using few-shot examples:

$$\mathbf{w}_{\text{new}} = \arg\min_{\mathbf{w} \in \Delta^{B-1}} \sum_{(x, y_c, y_r) \in \mathcal{D}_i} \ell\left(\sum_{j \in [B]} w^{(j)} \left(\beta \log \frac{\pi_{\theta_j}(y_c|x)}{\pi_{\text{ref}}(y_c|x)} - \beta \log \frac{\pi_{\theta_j}(y_r|x)}{\pi_{\text{ref}}(y_r|x)}\right)\right). \tag{19}$$

## B   Additional Details on Experiments

**Evaluation Metrics** For any user $i \in \mathcal{U}_{\text{seen}} \cup \mathcal{U}_{\text{unseen}}$ (seen or unseen) we evaluate the performance of the learned reward basis $\mathbf{R}_\phi$ and preferences $\mathbf{w}_i$, on their unseen pairs of responses $\tilde{\mathcal{D}}_i$ (ie. data not present in $\mathcal{D}_{\text{train}}$ or $\mathcal{D}_{\text{fewshot}}$) as the fraction of responses the learnt reward model classified correctly, defined formally as:

$$\frac{1}{|\tilde{\mathcal{D}}_i|} \sum_{(x, y_c, y_r) \in \tilde{\mathcal{D}}_i} \mathbb{I}[\mathbf{w}_i^\top (\mathbf{R}_\phi(x, y_c) - \mathbf{R}_\phi(x, y_r)) > 0]. \tag{20}$$

To test how well the learnt parameters generalize, we consider the following:

1. **Generalizing to Unseen Pairs of Responses for Seen Users (Seen Accuracy):** This type of generalization involves predicting well for new pairs of responses for users whose preferences have already been learned from the training data. We do so by evaluating the learnt model's ($\phi, \{\mathbf{w}_i\}_{i \in \mathcal{U}_{\text{seen}}}$ from Eq. (10)) classification accuracy (via Eq. (20)) on $\mathcal{D}_{\text{test}}^{\text{seen}}$ which contains the seen users' labels on prompts and response pairs that are not present in the training dataset $\mathcal{D}_{\text{train}}$.

2. **Generalizing to New Users (Unseen Accuracy):** This type of generalization involves predicting well for users whose data was not part of the training data $\mathcal{D}_{\text{train}}$, i.e., completely new users with unseen preferences. These users are termed as **unseen users** $\mathcal{U}_{\text{unseen}}$. These users come with few labelled data points, that is termed as $\mathcal{D}_{\text{fewshot}}$, and this is used to few-shot learn these users' preferences $\{\mathbf{w}_i\}_{i \in \mathcal{U}_{\text{unseen}}}$ by optimizing Eq. (11), keeping $\phi$ fixed. We evaluate the learnt preference weights' accuracy (via Eq. (20)) on $\mathcal{D}_{\text{test}}^{\text{unseen}}$ which contains the unseen users' labels on prompts and response pairs that are not present in the training dataset $\mathcal{D}_{\text{fewshot}}$.

3. **Performance as the number of dialogs increases for few-shot estimation:** It is natural that a new user gradually builds up feedback data on a LLM server. So far we were working in the setup where the LLM has already collected some feedback for the user based on some conversations, and see how the estimated preferences generalizes to future conversations. We consider how the performance varies as the LLM provider builds up multiple conversations with users. We do so by increasing the dataset size for $\mathcal{D}_i \in \mathcal{D}_{\text{fewshot}}$ while learning $\mathbf{w}_i$ via Eq. (11) for all $i \in \mathcal{U}_{\text{unseen}}$.

### B.1   Architecture of Baselines and Training Hyperparameters

**Functional Form of Reward Basis Function (LoRe):** For all our experiments, we use a pre-trained reward model (Liu et al., 2024) to output embeddings of dimension $D = 4096$, which we denote as $\mathbf{e} : \mathcal{X} \times \mathcal{Y} \mapsto \mathbb{R}^D$. We keep this embedding function frozen and learn a linear transformation on top of these fixed embeddings. The reward basis function is thus defined as:

$$\mathbf{R}_\phi(x, y) = \mathbf{A}\mathbf{e}(x, y), \tag{21}$$

where $\mathbf{A} \in \mathbb{R}^{B \times D}$ is a learnable matrix, representing the linear transformation applied to the pre-trained embeddings. By keeping the embedding function fixed and only learning the linear transformation, we can effectively adapt the pre-trained reward model to user preferences from specific datasets while leveraging its rich feature representations.

**Hyperparameters: A** is learned by using Adam (Kingma, 2014) on Eq. (10) with learning rate 0.5. For few-shot adaptation, we use Adam with learning rate 0.1 on Eq. (11).

The number of basis $B$ is selected from $\{2, 5, \ldots, 50\}$ through cross validation on a held out validation set.

**Experimental Setup for baselines:** We follow the exact training code, model architecture, and hyperparameters as described in the original implementations of PAL (Chen et al., 2024a) and VPL (Poddar et al., 2024), and run them on our benchmark datasets without modification. Detailed hyperparameter settings for PAL and VPL are listed in Table 4 and Table 5, respectively.

Table 4: The training hyperparameter setting of PAL (Chen et al., 2024a).

| Hyperparameters | Values |
|---|---|
| B (Number of prototypes) | selected through cross validation from $\{2, 5, 10, \ldots, 50\}$ |
| Batch size | 4 |
| Projectors | mlp-2layer-geul-dropout0 |
| Learning rate of projectors | 1e-4 |
| Learning rate of user weights | 5e-3 |
| Weight decay of projectors | 0.01 |
| Weight decay of user weights | 0.0 |
| Dimension of preference embedding | 512 |

Table 5: Hyperparameters for VPL (Poddar et al., 2024).

| Hyperparameter | Value |
|---|---|
| Pair Encoder Architecture | 2 layer MLP with LeakyReLU |
| Hidden Dimension | 512 |
| Latent Dimension | 512 |
| Learning rate | $1.0000 \times 10^{-4}$ |
| Learning rate scheduler | Cosine with 3% warmup steps |
| Batch size | 32 |
| Optimizer | AdamW (with weight decay = 0.001) |

**Parameter Efficiency of LoRe** As shown in Table 6 and Figure 3, LoRe is significantly more lightweight in terms of total trainable parameters compared to PAL and VPL. While PAL and VPL rely on large MLP architectures and per-user latent representations, LoRe uses only a simple linear projection on frozen embeddings, combined with a small set of basis-user interactions. This design leads to a much more compact model, especially as the number of users increases, enabling scalable personalization without compromising effectiveness.

Table 6: Scaling of Total Trainable Parameters for VPL, PAL, and LoRe. Here, $N$ is the number of seen users and $B$ refers to the number of prototypes in PAL, and number of basis (rank) in LoRe.

| Method | Architecture Details | Parameter Count |
|---|---|---|
| **VPL** | 2 layer MLP on frozen embeddings | $4096 \times 512 + 512 \times 512 + 512$ |
| **PAL** | 2 layer MLP on frozen embeddings (B prototypes) | $4096 \times 512 + 512 \times 512 + 4096 \times B + B \times N$ |
| **LoRe** | Linear Transformation on frozen embeddings (B basis) | $B \times 4096 + B \times N$ |

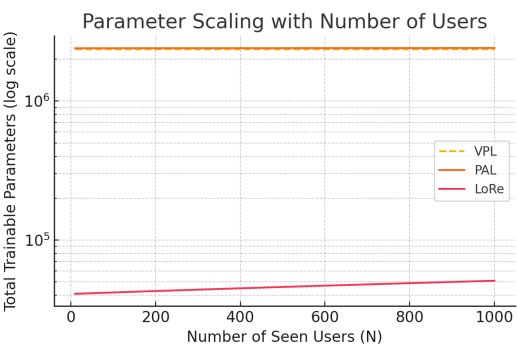

Figure 3: Trainable parameter count vs. number of seen users (log scale). LoRe scales significantly more efficiently than PAL and VPL as the number of users increases. Unlike VPL and PAL, which rely on large MLPs and high-dimensional prototype representations, LoRe uses a lightweight linear projection with shared basis vectors, resulting in dramatically fewer parameters while retaining personalization capabilities.

