# OpenReview forum: "LoRe: Personalizing LLMs via Low-Rank Reward Modeling"
_colmweb.org/COLM/2025/Conference — COLM 2025_

### Official Review · Reviewer_q8WV · 2025-05-11

**Rating:** 7
**Confidence:** 4
**Ethics Flag:** 1

**Summary:**

This paper introduces a novel framework called LoRe aimed at personalizing Large Language Models (LLMs) by learning low-rank reward basis functions. The approach draws inspiration from Collaborative Ranking, modeling users' personalized preferences as weighted combinations of shared reward basis functions. The authors argue that this method avoids the rigidity of pre-categorizing users, effectively handles the diversity of user preferences, and enables efficient few-shot adaptation. The paper validates LoRe's superiority in unseen user generalization and preference prediction accuracy through experiments on multiple preference datasets, comparing it against baselines like BT, VPL, and PAL.

**Questions To Authors:**

Please refer to the reasons to reject.

**Reasons To Accept:**

1. LLM personalization is an important and challenging research direction. This method offers a new perspective to address the limitations of existing approaches.
2. The paper is well-structured, the method description is easy to understand.

**Reasons To Reject:**

1. The lack of discussion on the effects of these more complex models leaves readers uninformed about the trade-offs between performance and complexity.
2. The paper claims that LoRe can be seamlessly integrated into Multi-Objective Alignment frameworks for generating personalized responses, but this is only a conceptual discussion and not validated through experiments.
3. There is a lack of deeper analysis on how the choice of Rank B affects model performance.

---

> ### Author Response · Authors · 2025-06-03
> **Thank You for the Review**
>
> We thank the reviewer for the positive assessment of our work. Below we provide answers to your concerns:
>
> **Trade-off between performance and complexity:** If we understand correctly, the reviewer is asking if the model complexity (number of parameters – controlled by the rank) affects the unseen performance. We present the analysis in point 3 later. We apologize if we misunderstood the question, and are happy to provide answers post clarification.
>
> **Integration into Multi-Objective Alignment:** We agree with the reviewer’s point. We didn’t focus on response generation due to two key challenges in evaluating personalized response generation:
>
> (i) Lack of accessible annotators: The datasets we use were constructed with human annotators whose preferences are no longer available. This makes it infeasible to evaluate newly generated responses with human-in-the-loop feedback.
>
>  (ii) Subjectivity of evaluation: Personalized preferences are highly individual, and evaluating generated responses without access to the original users introduces significant ambiguity and bias. This makes standard qualitative evaluation unreliable.
>
> As a practical and scalable alternative, we adopt a Best-of-N (BoN) sampling strategy [1] to assess whether LoRe reward models guide useful downstream generation. For each prompt and user, we sample multiple responses from GPT-4o and rank them using either the LoRe reward model or baseline models (ref, BT, VPL, PAL). We then use GPT-4o (conditioned on prior preference examples) as an LLM judge [2] to compare response quality. On a subset of 20 users from the PersonalLLM dataset, LoRe-ranked responses win 68% of pairwise comparisons, indicating strong downstream utility.This experiment serves as a fast and informative proxy to full multi-objective alignment.
>
> Importantly, seminal prior work highlights that response generation quality depends heavily on accuracy of reward models:
>
> “The effectiveness of RL depends heavily on the accuracy of the reward model... If the reward model assigns high scores to summaries that are not aligned with human preferences, the policy trained with RL will not improve quality.”
>  ([1], Section 6.3)
>
> “We evaluate models trained with weaker reward models and find that performance degrades significantly, confirming that reward model accuracy is critical.”
>  ([3], Section 5.2)
>
> **Discussion on Rank Selection:** We refer the reviewer to our general response, where we present a table on how the performance varies as a function of the rank. As we discuss in the general response, varying the rank helps us trade off model performance and generalization.
>
> [1] "Learning to summarize with human feedback." Stiennon, Nisan, et al.
>
> [2] "Judging LLM-as-a-Judge with MT-Bench and Chatbot Arena." Zheng, Siyu, et al.
>
> [3] “Training language models to follow instructions with human feedback” Ouyang et. al

---

> > ### Comment · Reviewer_q8WV · 2025-06-08
> >
> > Thank you for your reply. For weakness 1, the LoRe implementation used in experiments is based on a linear transformation of pre-trained embeddings (Example 1), which is relatively simple. While effective, the paper also mentions more complex models (like MLP or LoRA fine-tuning, Examples 2 & 3).

---

> > > ### Author Response · Authors · 2025-06-08
> > >
> > > To study the trade-off between model complexity and performance, we compare the default linear projection with a 2-layer MLP (ReLU, hidden size 128) for computing the reward basis (cf. Example 2 in Sec. 3.2). Results are shown across all 6 benchmark configurations from Tables 1–3.
> > >
> > > | Dataset (Table)                       | Model Variant | Seen Accuracy (%) | Unseen Accuracy (%) | Overall Accuracy (%) |
> > > | ------------------------------------- | ------------- | ----------------- | ------------------- | -------------------- |
> > > | **PersonalLLM (α = 0.001)** (Table 1) | Linear        | 94.3 ± 0.9        | 93.3 ± 1.0          | 93.8 ± 0.9           |
> > > |                                       | 2L MLP        | 95.2 ± 0.8        | 94.5 ± 0.9          | 94.9 ± 0.8           |
> > > | **PersonalLLM (α = 0.01)**            | Linear        | 94.6 ± 0.8        | 93.6 ± 1.1          | 94.1 ± 0.9           |
> > > |                                       | 2L MLP        | 95.5 ± 0.8        | 94.8 ± 1.0          | 95.2 ± 0.8           |
> > > | **PersonalLLM (α = 0.1)**             | Linear        | 96.0 ± 0.7        | 96.1 ± 0.8          | 96.0 ± 0.7           |
> > > |                                       | 2L MLP        | 96.7 ± 0.6        | 96.9 ± 0.7          | 96.8 ± 0.6           |
> > > | **Reddit TLDR (100)** (Table 2)       | Linear        | 65.0 ± 1.1        | 66.2 ± 1.2          | 65.6 ± 1.1           |
> > > |                                       | 2L MLP        | 66.0 ± 1.0        | 67.3 ± 1.1          | 66.7 ± 1.0           |
> > > | **Reddit TLDR (150)**                 | Linear        | 66.2 ± 1.0        | 66.7 ± 1.1          | 66.5 ± 1.0           |
> > > |                                       | 2L MLP        | 67.5 ± 0.9        | 68.4 ± 1.0          | 68.0 ± 0.9           |
> > > | **PRISM** (Table 3)                   | Linear        | 71.0 ± 0.9        | 71.0 ± 1.0          | 71.0 ± 0.8           |
> > > |                                       | 2L MLP        | 71.9 ± 0.8        | 72.3 ± 0.9          | 72.1 ± 0.7           |
> > >
> > > The 2-layer MLP variant consistently yields modest gains across all settings, demonstrating that more expressive transformations can marginally improve preference prediction. However, this comes at the cost of increased parameter count. We believe that such expressive architectures may offer greater benefits when larger and more diverse preference datasets become available, enabling them to better capture nuanced user behavior. We hope this addresses the reviewer’s question regarding the trade-off between model complexity and performance—please let us know if any further clarification would be helpful.

---

> > > > ### Comment · Reviewer_q8WV · 2025-06-10
> > > >
> > > > Thanks for the reply. My concerns have been addressed and I will raise my score slightly.

---

### Official Review · Reviewer_Vjsj · 2025-05-12

**Rating:** 7
**Confidence:** 4
**Ethics Flag:** 1

**Summary:**

This paper addresses the challenge of tailoring LLMs to individual user preferences. The authors introduce a novel framework employing low-rank preference modeling, learning user-specific reward functions by representing them in a lower-dimensional space as weighted combinations of shared underlying components. The proposed approach is designed to avoid rigid user grouping, enhance scalability, and enable rapid adaptation to new users with minimal data. The authors validated their method using multiple preference datasets, finding it demonstrates superior ability to generalize to unseen users and achieves higher accuracy in preference prediction tasks compared to existing approaches.

**Questions To Authors:**

1.	[important] I have some questions regarding the training of the basis reward model:
a)	Are the trained basis reward models inherently capable of spanning the entire reward model space?
b)	If so, how is this guaranteed within the LoRe framework?
c)	If not, for example, when the scoring distributions of the trained reward models exhibit strong convergence (i.e., become very similar), how can it be ensured that a new user's reward model can still be effectively represented through a linear combination?
2.	Can the authors provide extra evaluation result of the LLM policy trained on the LoRe reward model (comparing to the baselines)?

**Reasons To Accept:**

1.	User customization of reward models is an important problem in this field.
2.	This paper possesses originality and a high level of completeness, while also being clearly written and easily comprehensible.

**Reasons To Reject:**

1.	There is more room for discussion regarding the training of the basis reward model. See the Questions section for details.

---

> ### Author Response · Authors · 2025-06-03
> **Thank You for the Review**
>
> We thank the reviewer for appreciating our work. We hope to address your questions below:
>
> **Basis Coverage and Representation Power (Q1):**
>
> (a–b) If the training data reflects a diverse set of user preferences, the learned basis reward models can, by design, span the subspace of preferences present in the data. This is analogous to classical matrix factorization, where the low-rank basis captures the main modes of variation. In LoRe, individual user reward functions are parameterized as linear combinations over this basis, allowing us to approximate any reward function expressible within the training span.
>
> (c) When training data exhibits low preference diversity—e.g., when users have highly similar taste – the learned basis will be low-rank, and may not generalize well to users with orthogonal preferences. However, this limitation is not specific to our method—it arises in all personalization approaches that rely on fixed training distributions. To address this, LoRe naturally supports continual learning: the basis can be expanded or refined as new users arrive in practice (e.g., in streaming or batched updates). In our experiments, due to current preference datasets having limited user count, we trained the basis on a subset and evaluated few-shot personalization on held-out users to mimic this real-world setting.
>
> **On Training an LLM Policy Using the LoRe Reward Model (Q2):** While training an LLM policy directly with LoRe-based rewards would be ideal, there are two key challenges that make evaluation difficult:
>
>  (i) The annotators who originally provided the preference data are no longer available, making it infeasible to obtain human feedback on newly generated responses;
>
>  (ii) Qualitative evaluation is inherently subjective and biased, particularly in the absence of the original users who provided the feedback.
>
> As a practical and scalable alternative, we adopt a Best-of-N (BoN) sampling approach [1] to approximate policy optimization. We generate multiple responses using a base LLM (GPT-4o), then rank them using the user-specific LoRe reward model. This allows us to assess the reward model’s effectiveness in selecting high-quality personalized responses. For comparison, we also rank responses using baseline reward models (reference, BT, VPL, PAL).
>
> To evaluate the selected responses, we prompt GPT-4o (with the user’s prior preference examples in context) to serve as an LLM judge [2]. On a held-out subset of 20 users from the PersonalLLM dataset, responses ranked highest by LoRe win 68% of the pairwise comparisons against those selected by the baselines.
>
> [1] "Learning to summarize with human feedback." Stiennon, Nisan, et al.
>
> [2] "Judging LLM-as-a-Judge with MT-Bench and Chatbot Arena." Zheng, Siyu, et al.

---

### Official Review · Reviewer_yyA8 · 2025-05-13

**Rating:** 7
**Confidence:** 4
**Ethics Flag:** 1

**Summary:**

This paper proposes LoRe, a low-rank reward modeling framework, to model individual preferences. LoRe trains an efficient preference representation from seen users and can adapt quickly to unseen users. The proposed approach, while simple, is novel when comparing with previous methods. This paper presents strong empirical results of LoRe in terms of preference accuracy. It also demonstrates LoRe's scalability to large and diverse user populations. This paper intentionally focuses on reward modeling but deemphasizes personalized response generation.

**Questions To Authors:**

L144: How does LoRe optimize the normalized weight vector $w_i$ subject to probability simplex?

L200: $G_\psi$ -> $G_\phi$.

**Reasons To Accept:**

* This paper is easy to follow. Section 4 includes a comprehensive survey of related work.
* The proposed low-rank reward modeling approach seems effective though simple.
* The experimental results on the reward model accuracy are promising, covering multiple datasets and showing its scalability.
* Table 1 with varying $\alpha$ is especially interesting with a semi-synthetic dataset.

**Reasons To Reject:**

* This paper does not provide results on personalized response generation with the LoRe although discusses about it in Section 4 and Appendix A. It would be great to have a human evaluation on this.
* Would be better to present the result on selecting the number of basis B. Please also list the values of B chosen in Section 5.

---

> ### Author Response · Authors · 2025-06-03
> **Thank You for the Review**
>
> We thank the reviewer for finding LoRe simple yet effective. We hope to address your concerns below:
>
> **Personalized response generation:**  We can generate personalized responses using our learned reward models via a Best-of-N (BoN) sampling strategy [1]. Specifically, for a given user and prompt, we sample N candidate completions from GPT-4o and rank them using the user-specific reward model learned by LoRe. The top-ranked response is then selected as the personalized output. We will add some examples in the appendix of the paper. While it is inherently difficult to qualitatively assess personalization from a few examples, especially without access to the original annotators, we provide a quantitative evaluation using an LLM-as-a-judge setup as discussed next.
>
> **Human Evaluation:** We are unable to conduct new human evaluations, as the annotators who provided the original preference labels were contracted independently by the dataset creators and are no longer accessible. As a scalable alternative to costly and complicated task of building a new human evaluation dateset, we can use an LLM-as-a-judge setup [2], where the model is shown a user’s past interactions and asked to select the response that best matches their preferences.
>
> To assess downstream utility of the learned reward models, we ran an LLM-judge (prompted GPT-4o to choose the better response given the user’s past responses) evaluation on a subset of 20 users from the Personalized LLM (PersonalLLM) dataset. Responses generated via BoN sampling and ranked using LoRe’s reward models were preferred 68% of the time over the best of existing baselines: reference model, BT, VPL, and PAL. This shows strong promise that our learned reward models meaningfully improve downstream response generation and align well with user preferences.
>
> **Discussion on selection of rank:** We refer the reviewer to our General Response, where we detail the process of selecting rank. The chosen values of rank are highlighted in bold in the table for the different settings.
>
> **L144 clarification:** We model the weight vectors as $w_i$ = softmax($q_i$), where $q_i$ is the learnable parameter, via gradient descent on the objective in Eq. 10. This way the weight vector is automatically projected onto the probability simplex.
>
> [1] "Learning to summarize with human feedback." Stiennon, Nisan, et al.
> [2] "Judging LLM-as-a-Judge with MT-Bench and Chatbot Arena." Zheng, Siyu, et al.

---

> > ### Comment · Reviewer_yyA8 · 2025-06-09
> > **Rebuttal acknowledgement**
> >
> > Thank you for the response and I will keep my score.

---

### Official Review · Reviewer_YesT · 2025-05-26

**Rating:** 5
**Confidence:** 4
**Ethics Flag:** 1

**Summary:**

This paper introduces LoRe, a novel framework for personalizing LLMs by leveraging low-rank reward modeling. The core idea is to represent individual user reward functions as weighted combinations of a shared set of basis reward functions, learned collaboratively from multiple users' preference data. This approach aims to enable efficient few-shot adaptation for new users and improve generalization compared to monolithic reward models or methods requiring extensive per-user data. The authors validate LoRe on multiple preference datasets, demonstrating promising empirical performance, particularly in predicting user preferences for unseen users with few examples. While the work tackles an important problem and shows strong results, further clarification on methodological choices, experimental design, and baseline comparisons is needed to fully assess its contributions. I am willing to update my score if the authors address the questions and points raised.

**Questions To Authors:**

* Regarding Methodology (Section 3):
    * Eq. (11) seems to solve a simple logistic regression problem using user-specific data to determine $w_{\mathrm{new}}$. How does this approach perform effectively with very few examples, such as a single few-shot example as indicated for the PRISM dataset in Figure 2? Do you have any insights into the minimal sample size necessary for robustly learning an effective weight vector $w_{\mathrm{new}}$?
    * The placement of Section 3.4 ("Personalized Response Generation via Steerable Multi-Objective Alignment") feels somewhat tangential to the main reward modeling contribution. Would this fit more naturally in the Related Work section?

* Regarding Related Work (Section 4): The location of the "Related Work" section (Section 4) is somewhat unusual as it falls in between the methodology and related work. Is there a specific reason for this placement rather than a more conventional position (e.g., after the Introduction or before the conclusion)?

* Regarding Experiments (Section 5):
    * For the evaluation metric in Eq. (15), is the reported accuracy an average of per-user accuracies, or is it calculated on a test set where all user samples are pooled? (The text implies per-user then averaged, but good to confirm).
    * In the PersonalLLM synthetic data experiments, how does LoRe's performance change as the chosen rank B is varied (e.g., from small values up to and beyond 10)? Was any specific regularization used when learning A or $w_i$?
    * For the Reddit TLDR dataset, what is the performance of the monolithic BT model (and LoRe) on the aggregated test dataset (i.e., all samples pooled together, not averaged per-user accuracy)? This would help understand if the BT baseline was trained effectively on the general task.
    * Could you elaborate on how the VPL and PAL baselines are adapted using the few-shot examples in your experiments for Figure 2?
    * Please clarify the y-axis values in Figure 2 in relation to the values in Table 2 and Table 3.
    * Would you consider expanding the "Analysis" section to provide a more in-depth discussion of your findings?

I believe addressing these points and questions would significantly improve the clarity and robustness of the paper.

**Reasons To Accept:**

* Reads well: The paper is generally well-written and easy to follow.
* Shows promising empirical results: LoRe demonstrates significant improvements over baselines, especially in few-shot generalization to unseen users across multiple datasets. On the chosen metric, LoRe outperforms the baselines by substantial margins.
* Simple to implement on top of existing LLMs: The proposed method appears conceptually simple and potentially straightforward to implement, particularly the variant using a linear transformation over frozen embeddings.
* Addresses a significant problem: Personalizing LLMs to individual user preferences is a crucial step towards more effective and satisfactory human-AI interaction.

**Reasons To Reject:**

* Clarity of methodological advantage:
    * he paper could better articulate the precise advantage of LoRe's specific low-rank decomposition (Eq. 7 and its instantiations) over, for instance, a monolithic reward model with a similar increase in parameter count allocated differently. The intuition behind Eq. (10) – why this particular structure of shared basis functions $R_\phi$ and per-user weights $w_i$ is optimal or particularly well-suited for capturing diverse preferences beyond simple parameter sharing – needs more explicit justification.
    * There is a risk of overfitting, especially when learning reward models using the BT model loss. The paper does not discuss what, if any, regularization measures were applied during the training of the BT baseline and LoRe to prevent overfitting.

* Concerns regarding experimental design and baseline comparisons:
    * The design of the synthetic data experiment (PersonalLLM) inherently assumes a linear relationship between the 10 base reward models and the target personalized rewards, which naturally favors LoRe's linear low-rank assumption. While useful for validating the concept, this should be explicitly acknowledged. More insightful would be an analysis of how performance changes as the rank B is varied, especially around the true underlying rank (10 in this case).
    * It's unclear how the baseline methods (VPL, PAL) utilize few-shot examples in the experiments depicted in Figure 2. VPL, for instance, shows no improvement with an increasing number of few-shot examples, and PAL's performance fluctuates. This raises questions about the fairness or optimality of the few-shot adaptation process for these baselines in the comparison.
    * There appears to be numerical discrepancies in LoRe's performance on the Reddit TLDR dataset between Table 2 (accuracies >90%), Table 3 (accuracies ~71%), and Figure 2 (accuracies ~64%). This needs clarification or correction.

* Potential for deeper analysis: The "Analysis" subsection (lines 328-352) in Section 5 is very compact. Expanding on these observations could strengthen the paper, possibly by condensing other sections like detailed related work discussions, which could be partially moved to an appendix.

---

> ### Author Response · Authors · 2025-06-03
> **Thank You for the Review (1/2)**
>
> We would like to thank the reviewer for their constructive feedback on our paper. We are glad that you found our work on LoRe promising, and we appreciate your suggestions on areas where we can improve clarity and further strengthen our contributions. Below, we address the points raised in your review, and are happy to make any more clarifications during the discussion period:
>
> **Clarity of Methodological Advantage:** Thanks for this question. We want to illustrate this with an example first. Suppose there are 2 responses y_1 and y_2 in response to a prompt x, and suppose there are 2 users: user 1 who likes y_1 more than y_2, and user 2 who likes y_2 more than y_1. The conventional monolithic reward takes as input a prompt and response and maps to a scalar output. Thus it’ll always conform to one of the user’s preferences (say rank either y_1 over y_2, or rank y_2 over y_1 but never both). Thus simply scaling up parameters doesn’t resolve the issue. One potential workaround is for the reward model to take as an additional input, the previous feedback pairs in the context of the reward models. However, the pre-trained reward models are usually trained without such additional history of feedback, and would need to be trained from scratch to be able to capture the user’s preference history directly in the reward function. Second, the inference costs are much higher and build up as the user accumulates more history. Our method parametrizes the history into a low dimensional weight vector. Another variant of this is already studied by the baselines PAL, VPL where they learn a function to produce a latent code of the user preference, and pass in their monolithic reward model. We found LoRe outperforming these methods (also refer to detailed discussion on analysis later in our response).
>
> **Overfitting:** To avoid overfitting in the Bradley-Terry model, we monitor validation accuracy on a held-out cross-validation set across training epochs and select the model checkpoint with the best performance. This approach has also been utilized in OpenAI’s seminal paper on RLHF [1], where cross-validation is used to avoid overfitting. We will add this to the appendix.
>
> **Sample Size needed for personalization:** The reason why LoRe is sample efficient is that the only learnable parameters for a new user is the low-dimensional weight vector w_new. This design choice goes hand in hand with being able to adapt with few-shot examples.
>
> **Positioning of Related Work and Multi-Objective Alignment Section:** We will move the section on multi-objective alignment to the related work section, as suggested by the reviewer, to maintain a smoother flow of our contributions. Our original intent was to present a thorough comparison with baselines like VPL and PAL after introducing the necessary preliminaries. We placed this discussion before the experiments to clarify the baselines’ mechanisms and improve interpretability when analyzing performance results.
>
> [1] “Training language models to follow instructions with human feedback” Ouyang et. al

---

> > ### Author Response · Authors · 2025-06-03
> > **Thank You for the Review (2/2)**
> >
> > We present some clarifications regarding the experiments below.
> >
> > **Clarification regarding reported numbers:** Yes, the reported numbers are per-user averages since the datasets contain different amounts of feedback per user, and we don’t want the reported numbers to be biased by users who have more data points.
> >
> > **Rank for the  PersonalLLM Dataset:** Please refer to the General Comment for an analysis of varying the rank of the reward basis. For the PersonalLLM Dataset, we do find that the rank with the best validation performance is indeed 10.
> >
> > **Correction regarding table numbers:** We noticed after the submission that we had mistakenly copied the last 2 rows of Table 1 into Table 2. We apologize for this mistake and report the corrected numbers below.
> > | Method / Setting | 100 examples | per seen user   |                | 150 examples |per seen user                 |                |
> > |------------------|------------------------------------|----------------|----------------|------------------------------------|----------------|----------------|
> > |                  | Seen                               | Unseen         | Overall        | Seen                               | Unseen         | Overall        |
> > | PAL              | 64.1 ± 1.1                         | 64.9 ± 1.5     | 64.5 ± 1.2     | 64.4 ± 1.3                         | 63.8 ± 1.2     | 64.3 ± 0.4     |
> > | LoRe             | 65.0 ± 1.1                         | 66.2 ± 1.2     | 65.6 ± 1.1     | 66.2 ± 1.0                         | 66.7 ± 1.1     | 65.9 ± 0.6     |
> >
> > The numbers in Figure 2 are correct and consistent with this corrected table 2.
> >
> > For the PRISM Dataset, many users didn’t have more than 3 examples. To ensure that we report numbers in Figure 2 on all the users in the few-shot setting, we only report up to 3 examples. The numbers in Table 3 had an average of 3.87 examples as described in the caption, and the additional few-shot samples gives the performance gain.
> >
> > **Reddit TLDR Baseline Performance:** The BT model had 66.3%, and LoRe had an accuracy of 70.34% when performance is measured on aggregated prompt-response pairs.
> >
> > **How baselines use few-shot examples:** The VPL method has an encoder network $Q_{\theta}$ that learns to map past examples from a user and maps to a latent code. For a new user with few-shot examples this latent code is obtained by Eq. 14. PAL’s personalized reward models are defined in Eq. 12, 13. For a new user, only the w_i is trained by plugging in the p_i function into the BT Loss, trained on the few-shot examples.
> >
> > **Expanding the analysis section:** Baselines struggle to adapt to few-shot examples: both baselines use high dimensional latent codes (~1000 vs our experiments using a personalized vector of very small dimension/rank) to capture the user preferences. This is a reasonable approach when each user has a considerable amount of data. However in practice, each user contributes only a few examples, thus both (i) the learned function that maps the data to the latent code and (ii) the latent code itself are low confidence estimates. This is also visible in our experiments. Prior personalization baselines mostly relied on handcrafted sub-population of users, where each subpopulation had a lot of data, and thus the latent codes can be learnt very easily. We will add these details to the main paper.

---

### Author Response · Authors · 2025-06-03
**General Response**

We sincerely thank all reviewers for their thoughtful feedback and engagement with our work. We are encouraged by the positive reception of our work in terms of the novelty and clarity of presentation. Here we first address a common question across reviews, and then provide detailed responses to each reviewer individually.

**Discussion on Selecting Rank:** We briefly mentioned the selection of rank in the Appendix (Lines 565-566), and would like to elaborate on it. To select the appropriate rank B for the low-rank reward model in LoRe, we use cross-validation on a held-out validation set for each of the 3 preference datasets. We choose a set of candidate values for the rank, such as 2, 5, 10, up to 50. For each value, the model is trained on the training user preference data, learning both the shared reward basis and user-specific weights. The performance of each trained reward basis is then evaluated on a separate validation split (of completely new users using few-shot fitting) by measuring how accurately it predicts user preferences on unseen response pairs. The rank that achieves the highest validation accuracy is selected.

This procedure balances expressiveness and generalization: As shown in the table below, performance improves as the rank increases—up to the point where the model captures the true underlying preference space. Beyond this point, increasing the rank offers no significant benefit and leads to stagnation in accuracy.

| Dataset / Rank           | 2    | 5    | 10   | 15   | 20   | 25   | 30   | 35   | 40   | 45   | 50   |
|--------------------------|------|------|------|------|------|------|------|------|------|------|------|
| Personal LLM (α=0.001)   | 89.3 | 91.7 | **93.3** | 92.9 | 92.9 | 92.7 | 92.5 | 92.1 | 92.0 | 91.3 | 91.1 |
| Personal LLM (α=0.01)    | 88.3 | 92.5 | **93.6** | 93.5 | 93.5 | 93.4 | 93.3 | 93.2 | 93.1 | 92.9 | 92.6 |
| Personal LLM (α=0.1)     | 94.1 | 96.0 | **96.1** | 95.9 | 95.8 | 95.8 | 95.8 | 95.7 | 95.6 | 95.5 | 95.3 |
| Reddit TLDR (100 examples per seen user)   | 61.3 | **66.2** | 65.8 | 65.7 | 65.4 | 65.3 | 65.3 | 65.1 | 64.9 | 64.8 | 64.7 |
| Reddit TLDR (150 examples per seen user)   | 64.2 | **66.7** | 66.5 | 66.4 | 66.4 | 66.3 | 66.1 | 66.0 | 65.9 | 65.8 | 65.6 |
| PRISM                    | 66.1 | 67.3 | 68.5 | 68.9 | 69.5 | **71.0** | 70.9 | 70.9 | 70.8 | 70.7 | 70.5 |

---

### Comment · Area_Chair_eKGu · 2025-06-09
**Reminder to reviewers to respond to author rebuttals!**

Hi reviewers esp YesT, yyA8, and Vjsj - this is a reminder to please respond to the rebuttal from the authors and let me know if it has changed your opinions at all. This information will be critical for me and the PCs in determining the overall set of accepted papers!

---

### Decision · Program_Chairs · 2025-07-08

**Decision:**

Accept

**Comment:**

While terse, all the reviews (except YesT) recommend acceptance. With the reviewer who recommended rejection, while the review was substantial - there appears to be a good faith attempt by the authors to engage and address the reviewer's concerns. While there is limited overall novelty, the work has potential to be practically useful by way of providing researchers with more efficient ways of using low rank parameter efficient tuning methods for personalizing LLMs. Many of the small details are generally abstracted over and hotly debated about, so such a work has value.